# Effectiveness of community-based health education and home support program to reduce blood pressure among patients with uncontrolled hypertension in Nepal: A cluster-randomized trial

**Mahesh Kumar Khanal**[1]*, **Pratiksha Bhandari**[2], **Raja Ram Dhungana**[3], **Pratik Bhandari**[4], **Lal B. Rawal**[5,6,7], **Yadav Gurung**[8], **K. N. Paudel**[9], **Amit Singh**[9], **Surya Devkota**[10], **Barbora de Courten**[11]

1 Provincial Ayurveda Hospital, Ministry of Health and Population, Lumbini Province, Nepal, 2 Rapti Life Care Hospital Pvt. Lt. Tulsipur, Dang, Nepal, 3 Institute for Health and Sport, Victoria University, Melbourne, Australia, 4 Faculty of SEBE, Deakin University, Warun Ponds, VIC, Australia, 5 School of Health, Medical and Applied Sciences, College of Science and Sustainability, Central Queensland University, Sydney Campus, Australia, 6 Physical Activity Research Group, Appleton Institute, Central Queensland University, Rockhampton, Australia, 7 Translational Health Research Institute (THRI), Western Sydney University, Penrith, Australia, 8 Child and Youth Health Research Center, Auckland University of Technology, Auckland, New Zealand, 9 Province Hospital, Ministry of Social Development, Karnali Province, Surkhet, Nepal, 10 Department of Cardiology, Manmohan Cardiothoracic Vascular and Transplant Centre, Institute of Medicine, Tribhuvan University, Kirtipur, Nepal, 11 Department of Medicine, School of Clinical Sciences at Monash Health, Monash University, Clayton, Victoria, Australia

* drmkkhanal@gmail.com

## Abstract

### Background

Hypertension is a major global public health problem. Elevated blood pressure can cause cardiovascular and kidney diseases. We assessed the effectiveness of health education sessions and home support programs in reducing blood pressure among patients with uncontrolled hypertension in a suburban community of Nepal.

### Methods

We conducted a community-based, open-level, parallel-group, cluster randomized controlled trial in Birendranagar municipality of Surkhet, Nepal. We randomly assigned four clusters (wards) into intervention and control arms. We provided four health education sessions, frequent home and usual care for intervention groups over six months. The participants of the control arm received only usual care from health facilities. The primary outcome of this study was the proportion of controlled systolic blood pressure (SBP). The analysis included all participants who completed follow-up at six months.

### Results

125 participants were assigned to either the intervention (n = 63) or the control (n = 62) group. Of them, 60 participants in each group completed six months follow-up. The

**Data Availability Statement:** All data files are available from the Dryad repository, available at https://doi.org/10.5061/dryad.7h44j0zts.

**Funding:** Nepal Family Development Foundation, Kathmandu, Nepal. The funders had no role in study design, data collection and analysis, decision to publish, or preparation of the manuscript.

**Competing interests:** The authors have declared that no competing interests exist.

proportion of controlled SBP was significantly higher among the intervention participants compared to the control (58.3% vs. 40%). Odds ratio of this was 2.1 with 95% CI: 1.01–4.35 (p = 0.046) and that of controlled diastolic blood pressure (DBP) was 1.31 (0.63–2.72) (p = 0.600). The mean change (follow-up minus baseline) in SBP was significantly higher in the intervention than in the usual care (-18.7 mmHg vs. -11.2 mmHg, p = 0.041). Such mean change of DBP was also higher in the intervention (-10.95 mmHg vs. -5.53 mmHg, p = 0.065). The knowledge score on hypertension improved by 2.38 (SD 2.4) in the intervention arm, which was significantly different from that of the control group, 0.13 (1.8) (p<0.001).

## Conclusions

Multiple health education sessions complemented by frequent household visits by health volunteers can effectively improve knowledge on hypertension and reduce blood pressure among uncontrolled hypertensive patients at the community level in Nepal.

## Trial registration

ClinicalTrial.gov: NCT02981251

## Introduction

Hypertension is one of the leading causes of cardiovascular diseases, kidney diseases, and premature death worldwide, increasingly affecting developing countries [1]. It was estimated that 1.13 billion people had hypertension in 2015 worldwide, the majority of them were in low-and middle-income countries [1]. In 2019, high SBP accounted for 10.8 million deaths [2]. Hypertension is responsible for 50% of deaths due to cardiovascular disease (CVD) worldwide [3], including 45% of death due to heart disease and 51% of death due to stroke [4].

The prevalence of hypertension is increasing in developing countries such as Nepal. From 2007 to 2013, the prevalence of hypertension increased from 18.1% to 25.2% [5, 6]. Similarly, a recent meta-analysis conducted by Dhungana et al. identifies an increase in hypertension by 6% over the past 20 years since the year 2000 [7]. Despite the increasing trend of hypertension, only about 4–8% of hypertensive patients have well-controlled blood pressure [5, 8]. Data revealed that blood pressure was normalized only in 3% of patients, over the last five years [7].

Evidence suggests that an alteration and adherence to smoking cessation, reduction to alcohol and salt intake, increased physical activity, and healthy nutrition intake with the abundance of fruits and vegetables [3] and the regular use of antihypertensive drugs can control blood pressure [9]. There are several interventions to improve patient's adherence to this healthy behavior. One of the effective strategies is behavior counselling or patient education that augments disease perception, motivation, and the ability to adhere to healthy habits [10]. A wealth of literature is available on the effectiveness of health education for blood pressure reduction among patients with hypertension [9, 11, 12]. Most of the studies are conducted in high-income countries. However, there is a paucity of information regarding the effect of health education on blood pressure in low-and middle-income countries like Nepal. Therefore, we aimed to assess the effectiveness of health education combined with frequent home visits in controlling blood pressure among uncontrolled hypertensive patients of the suburban community of Surkhet district of Nepal.

## Methods

### Study design and site

We conducted a community-based, parallel-group design, open-level, clustered randomized controlled trial in Birendranagar municipality of Surkhet district, Nepal. We considered a ward, the smallest administrative unit of the municipality, as a cluster and a unit of randomization. We used clustered randomization to avoid contamination between the groups. This study was conducted from January 2016 to August 2017. The intervention took place between February and July 2017.

The study site lies about 2500 feet above sea level and 600 kilometres west of the capital city of Nepal, Kathmandu. The total population of the city was 52,137 during the study and people were residing in 12 wards. All wards were inhabited by a multicultural, multiethnic, indigenous, and migrated population [13].

### Study participants

Initially, we conducted a cross-sectional study to screen the eligible participants from four clusters. The study method has been followed as described by Khanal et. al. [8]. This trial included people aged 30 years and above, those who had uncontrolled hypertension (SBP ≥140 mm Hg, and/or DBP ≥90 mm Hg) including participants on antihypertensive medication. The familiarity of Nepali language, availability of contact number, residing in the city for at least six months following the baseline data collection, and willingness to participate were other criteria of selection. We excluded the pregnant women, visually impaired or person with hearing loss, any bed-ridden patients, participants diagnosed with kidney disease, cancer, heart disease, chronic obstructive pulmonary disease, and mental illness.

### Randomization and study procedure

Initially, we identified eligible participants using a cluster sampling technique. Wards of municipalities were regarded as a cluster. From the twelve wards of the municipality, thirty percent of the wards (four wards) were chosen randomly. After the completion of participant recruitment, baseline data collection and obtaining informed consent in those four clusters, we performed randomization of clusters. A statistician, without previous knowledge of clusters, randomly assigned (1:1) four clusters to intervention and control groups, using computer-generated codes. We conducted a follow-up data collection after six months from patients of intervention and control groups using the same tool as in baseline. Separate enumerators involved in the final phase who were blinded to intervention and control group.

### Sample size

In our protocol, we aimed for a total sample size of 76 (38 from each cluster) in each arm of the trial with alfa = 0.05, beta = 0.20 and a non-response rate of 15% [14]. It was estimated based on the fact that normalized blood pressure can be achieved in 25% of uncontrolled hypertensive patients after health education while only 5.2% of patients achieve controlled blood pressure through usual care [15]. After the literature review, we used an inter-cluster correlation (ICC) = 0.01 for community-based hypertension trial to adjust for the design effect [16]. During the implementation of the research, we managed to randomize a total of 125 participants in either arm. As only 120 participants completed the trial, in the main analysis, we could analyze data of those who were followed up. However, we analyzed information of 125 samples in the sensitivity analysis by incorporating imputed missing values.

## Outcomes of the study

The primary outcome of the study was the proportion of hypertensive patients with normalized SBP in the intervention and the control group. The secondary outcomes were the proportion of controlled DBP, mean difference of SBP, DBP, body mass index (BMI), waist circumference (WC), and knowledge score after six months of intervention and usual care.

## Blood pressure measurements

We had provided robust training to enumerators to measure the blood pressure using an aneroid sphygmomanometer on the left arm. Based on the upper arm circumference of the participant, we used an appropriate cuff size. For accuracy, we had recorded two readings of every participant, first after 15 minutes of rest in the uncrossed leg and second after three minutes of the first measurement. The mean of two readings was used for the final analysis [17, 18].

## Hypertension control

Hypertension was defined as average SBP $\geq$ 140 mmHg and/or DBP $\geq$ 90 mmHg and/or history of taking antihypertensive medication in the last two weeks [17]. Controlled SBP was defined as SBP < 140 mmHg and controlled DBP was defined as DBP < 90 mmHg among hypertensive participants [19]. Finally, if both systolic and diastolic BP was below these levels, we defined them as controlled blood pressure.

## Variables related to behavior

We defined current smokers as those who were smoking cigarettes and who quit less than one month before the interview. Respondents who were chewing tobacco in the last 30 days were defined as current tobacco users [18]. We called respondents as current alcohol user if they had drunk alcohol within the last 30 days of data collection. We had shown a pictogram to estimate the number of servings of fruit and vegetable intake in a week. Fruit and vegetable intake of at least 400 grams or at least five servings per day was considered as sufficient fruit and vegetable consumption [18]. We used the Global Physical Activity Questionnaire (GPAQ), adapted from Stepwise Approach to Surveillance (STEPS survey) Nepal, to assess the physical activity level. We calculated the amount of physical activities in metabolic equivalents of task (MET) per minute per week based on STEPS questionnaires. We used a cut-off value of 600 MET-minutes per week or lower to define inadequate level (physically less active) [18].

## Anthropometric measurements

We used a portable stadiometer and weighing machine, respectively to measure height and weight. We used a tape to measure the waist circumference at the mid-point between the lower margin of the last palpable rib and the top of the iliac crest (hip bone) at the end of a normal expiration, holding the arms relaxed at the sides. The BMI was classified as underweight (<18.5 Kg/m$^2$), normal (18.5–24.9 Kg/m$^2$), overweight ($\geq$25 Kg/m$^2$), and obese ($\geq$30 Kg/m$^2$) [20]. Central obesity was defined as an increased waist circumference of more than 88 cm in women and more than 102 cm in men [21].

## Interventions

The participants in the intervention group received group-based interactive health education sessions, followed by home visits by community health volunteers, and locally available standard care. The detail of the interventions is provided as below:

**Interactive health education sessions.**    One medical graduate and one registered nurse facilitated four health education sessions during the six months intervention period. Those sessions were conducted at first, second, third and fifth months after the enrolment of the participants. In each session, about 15–20 participants attended the discussion for about two hours. Sessions were based on a specific syllabus (S1 Table). We designed the syllabus based on the patient information of the American Heart Association [22], the seventh report of the joint National Committee on Prevention, Detection, evaluation, and treatment of high blood pressure [17], the National Health Education Information and Communication Centre of Nepal [23], and World Health Organization (WHO) pocket guidelines for assessment and management of cardiovascular risk [24].

The health education syllabus is comprised of seven chapters: (1) communication, (2) introduction to hypertension, (3) causes/factors associated with hypertension and (4) complications of high blood pressure, (5) lifestyle management, (6) medication for hypertension and (7) WHO/ISH CVD risk prediction chart. In the first session, we discussed the previous knowledge, experiences, barriers, difficulties in changing behavior, taboos and misconception about high blood pressure to get the baseline knowledge and to make a comfortable environment among the participants and facilitator. Then, we focused on the meaning and measurement of blood pressure; grading, burden, causes, and complications of hypertension in detail, and a brief discussion on lifestyle management and antihypertensive medication. In the following sessions, at the outset, we summarized the previous health messages and discussed in detail lifestyle management and medication for high blood pressure.

In lifestyle management, we delivered the health information related to healthy eating, a nutritional recommendation based on Dietary approaches to stop hypertension (DASH), physical activity, maintaining a healthy weight, limit (or avoid) alcohol, quit tobacco, reducing stress, tips for changing daily life in the context of the living condition of the community. Moreover, the facilitators discussed indications of antihypertensive medication, types of medicines, mechanisms to reduce blood pressure, monitoring of blood pressure medication, ways to reduce side effects of medicine, importance and methods of adherence of medicine, and the WHO risk prediction chart.

**Home support through health volunteer.**    At least one community volunteer for each cluster was mobilized for motivating and supporting the participants in the community. We trained the health volunteer on accurate blood pressure measurement and behavior change counselling based on the same syllabus designed for the participants. The health volunteer visited the participant's home in 15–30 days intervals. The volunteers measured the blood pressure, discussed the problems facing personally or in the family in controlling blood pressure. They reminded about physical activities and fruit and vegetable intake, advised to visit their health professionals if blood pressure was not in the normal range, and provided feedback to the participants and their family members.

**Usual care for those in the control arm.**    The participants of the control group received locally available usual health care. During the time of the intervention, the Birendranagar municipality, Surkhet had one government regional tertiary level hospital, three fifteen-bedded private hospitals, one primary care level health centre, six polyclinics, and one Zonal Ayurveda Hospital. Further, the municipality had a district public health office and some ongoing public health-related programs within the municipality area.

## Randomization, study procedure, and follow-up

At first, we identified eligible participants using a cluster sampling technique. Among twelve wards, four wards were selected randomly. After the completion of participant recruitment

and baseline data collection in all four clusters, we randomly assigned two clusters in each arm; two with health education and home support and two with usual care groups. We conducted a follow-up data collection after six months from patients of intervention and control groups using the same tool as in baseline. Enumerators of the follow-up stage did not know about baseline information.

## Data collection

The first phase of the study was screening for an individual with high blood pressure in each cluster, the detail of this is present everywhere [8]. In brief, trained enumerators collected required data utilizing WHO STEPS questionnaires [18] to gather information related to age, sex, socio-economic status, smoking and tobacco use, alcohol, fruit, vegetable and salt intake, and physical activities. In addition, they recorded the previous diagnosis of hypertension, intake of any anti-hypertensive drugs within two weeks before the data collection. Additionally, we used another tool with ten questionnaires to assess hypertension-related knowledge, where six questionnaires had three multiple-choice answers and four had more than three multiple choice answers. Point one was given for correct answers, zero for incorrect and unanswered questions, whereas half point was allocated for partially correct answers. We developed questionnaires and adapted the scoring criteria based on the steps used by Chu-Hong Lu et al [15]. Before application, two independent translators had translated these tools into the Nepali language and one of the authors finalized the set of questionnaires. We had pre-tested the final versions of all tools among 20 participants before final implementation. These same tools were used for both baseline and follow-up data collection.

## Data quality and safety monitoring

We recruited a field supervisor to monitor the intervention. As proof of intervention, we used a register to record the date, time, activities and names of participants who were involved in the group counselling sessions and home visits. These activities were cross-verified by the field supervisor and were confirmed during the follow-up survey by asking participants. This supervisor was responsible for reviewing data quality and report any missing information and errors during data collection. To report any adverse effects of the intervention, we had provided the contact number of the field supervisor and principal investigator to the participants.

## Data analysis

We compiled, edited and entered data in Epidata 3.1. We removed duplicates and exported them to SPSS V.20.0 for final analysis. Complete case analysis as per protocol was done for the individual participants who completed the trial. Descriptive statistics (mean, standard deviation for continuous variables and proportion for discrete variables) were used to present the characteristics of the study subjects for both baseline and follow-up stages. Independent t-test (continuous variable) and Chi-square test (binary variable) were used to compare the difference between the groups. Additionally, we used Fischer's exact test if the expected frequency was small (less than five). A binary logistic regression model was applied with three separate dependent variables in the main analysis. First was presence or absence of normalized SBP at the follow-up. The second was presence or absence of SBP at the end of the trial and lastly as presence or absence of controlled blood pressure (both systolic and diastolic) at follow-up. Then, the models were adjusted for age, gender, level of education, ethnicity, clustering effect, baseline mean blood pressure (SBP for systolic, DBP for diastolic and both for blood pressure), and baseline knowledge score. Additionally, for mean SBP, DBP, knowledge score, BMI and waist circumference, we computed mean (SD) of change

(follow-up minus baseline). Box plot was also used to present effect of the intervention on mean SBP and DBP from baseline and follow up in both intervention and control groups. Additionally, we also performed a linear regression analysis of follow-up knowledge score using baseline knowledge score and intervention. In the end, we used Little's test for MCAR (missing completely at random) test of missing value to perform sensitivity analysis. We applied Expectation-Maximization (EM) imputation technique to impute missing data (5) using baseline and follow-up SBP and DBP of all available participants. All tests were two-tailed and $p < 0.05$ was considered statistically significant.

## Ethical consideration

We obtained ethical approval from the Ethical Review Committee of Nepal Health Research Council (Reg. no. 24/2016, on 17 February 2016). Every participant read the form and provided the written consent either by signature or with a thumb impression if participants could not write. We explained the study aims, data collection procedures, interventions stages and contents, benefits, risks, and confidentiality to every respondent before obtaining consent.

This trial was registered in the clinical trial registry (ClinicalTrial.gov: NCT02981251) on December 5, 2016.

## Results

### Participants' flow

In January 2017, we randomly assigned two clusters in the intervention group and two clusters in the control group. A number of participants who received interventions was 63 and those who were on usual care were 62. At six months of intervention, 3 participants either migrated, lost the phone or discontinued on the intervention group and 2 participants were lost to follow-up because of the same reasons in the control arm. As we did not have follow-up data of five participants, we analyzed 60 samples in both intervention and control groups for baseline and characteristics and intervention effect (Fig 1). In the end, we included all randomized participants in the sensitivity analysis. None of the participants reported experiencing any adverse effects due to intervention and nobody reported changes in medication use.

### Baseline socio-demographic characteristics of the participants

Forty-five percent were male in both control and intervention groups. The mean age was 56.6 years for both arms with standard deviation (SD) of 10.2 and 11.6 years, respectively. Majority in both groups were married, had no formal education and were above the poverty line. In the control group, 36.7% were employed, whereas 45% of the intervention group were household workers. All of the sociodemographic characteristics were similar as all of these were statistically non-significant (Table 1).

### Health behavior and drug intake of participants at baseline

Of the total, 21.7% and 13.3% of participants were current smoker in control and intervention group, respectively. Among these two groups, around one-fifth (21.7%) and one-third (36.7%) were consuming alcohol at the baseline. Most of participants did not consume adequate servings of fruits and vegetable (98% vs. 93%). Participants in the intervention group were less physically active (46.7%) compared to the usual care group (30%). Compared to the control group (53.3%), 40% of participants in the intervention group were taking antihypertensive medicine. All of these health behaviors were statistically non-significant (Table 2).

## CONSORT 2010 Flow Diagram

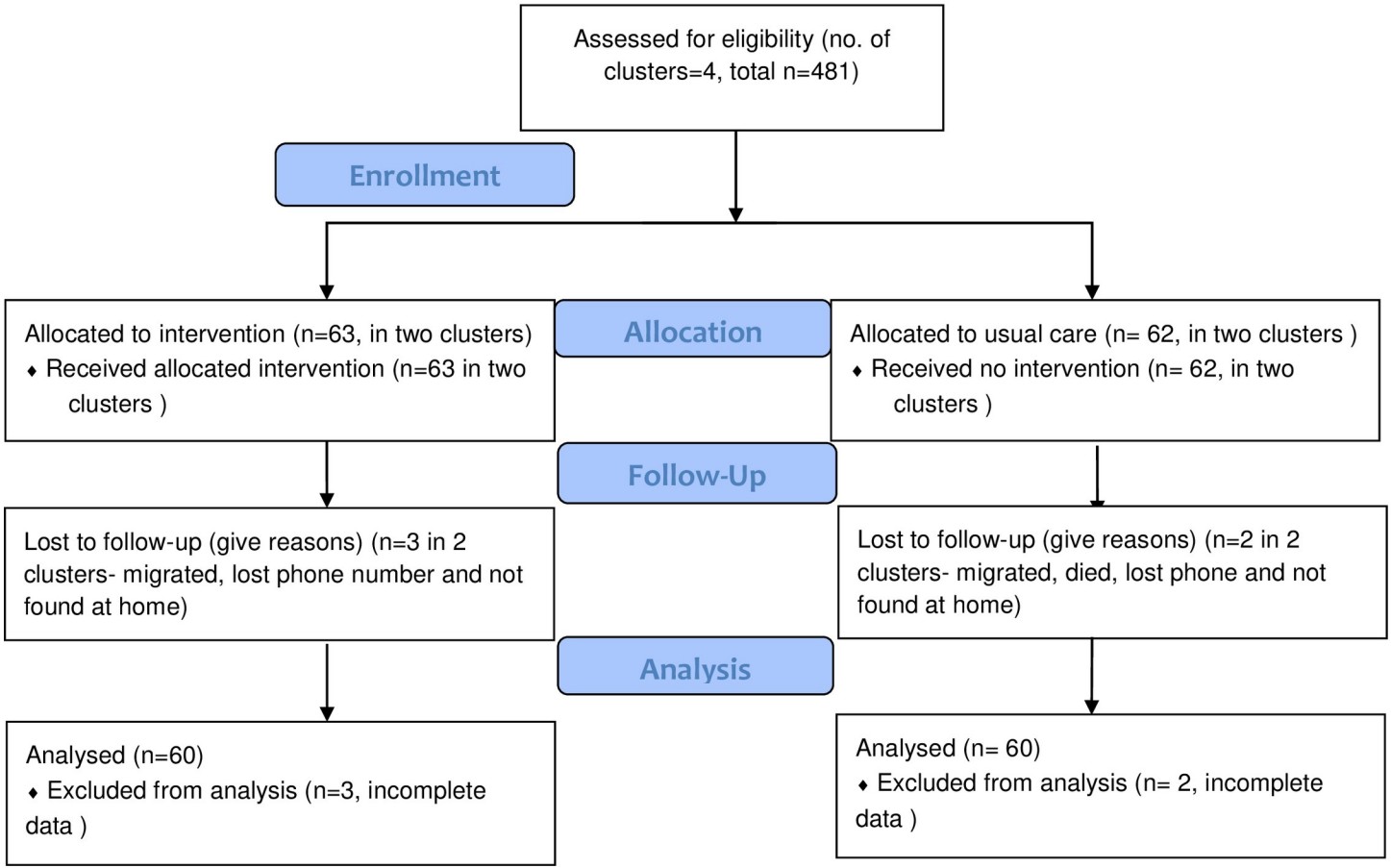

**Fig 1. CONSORT 2010 flow diagram of study participants.**

### Effectiveness of intervention

After six months of health education and community support, the proportion of participants with well-controlled SBP increased to 58.3% from 3.3% compared to only to 40% from 3.3% among the usual care group. The intervention had 2.1 times higher odds of having well-controlled blood pressure compared to usual care (p = 0.046), whose 95% confidence interval (CI) was 1.01–4.35. After adjusting for age, sex, ethnicity, education level, clustering effect, baseline knowledge score, the odds increased to 2.94 (CI 1.25–6.9). Similarly, percentage of the controlled DBP increased by 30% after the intervention compared to only 20% on usual care. The unadjusted odds ratio of controlled DBP was 1.31 (0.63–2.72) (p = 0.600). When we adjusted for age, sex, ethnicity, education level, clustering effect, baseline knowledge score, and baseline DBP, the odds ratio (OR) was 1.24 (0.55–2.80). Likewise, 38.3% of participants had well-controlled blood pressure six months after the intervention, whereas only 28.3% of respondents had controlled BP on usual care. The unadjusted odds ratio for controlled BP was 1.57 (0.73–3.38) and the adjusted OR was 1.79 (0.74–4.35) (Table 3).

The mean (SD) of SBP was 152.1 (12.9) mmHg and 156.6 (17.4) mmHg in control and intervention groups, respectively. At six months, the mean (SD) of SBP was 140.9 (19.5)

**Table 1. Baseline socio-demographic characteristics of study participants.**

| Variable | | Usual care (n = 60) | Intervention (n = 60) | P value |
|---|---|---|---|---|
| | | n (%) | n (%) | |
| **Sex** | | | | |
| | Male | 27 (45.0) | 27 (45.0) | 1.00 |
| | Female | 33 (55.0) | 33 (55.0) | |
| **Age (years)** | | | | |
| | 30–44 | 8 (13.3) | 9 (15.0) | 0.854 |
| | 45–59 | 26 (43.3) | 23 (38.3) | |
| | ≥ 60 | 26 (43.3) | 28 (46.7) | |
| | Mean (SD) | 56.6 (10.2) | 56.6 (11.6) | 0.972 |
| **Marital status** | | | | |
| | Unmarried | 1 (1.7) | 1 (1.7) | 0.424* |
| | Married | 53 (88.3) | 48 (8.0.0) | |
| | Single | 6 (10.0) | 11 (18.3) | |
| **Ethnicity** | | | | |
| | Brahman | 17 (28.3) | 8 (13.3) | 0.157 |
| | Kshetri | 11 (18.3) | 11 (18.3) | |
| | Shahi or Thakuri | 5 (8.3) | 6 (10.0) | |
| | Dalit | 16 (26.7) | 27 (45.0) | |
| | Janajati | 11 (18.3) | 8 (13.3) | |
| **Level of education** | | | | |
| | No formal education | 42 (70.0) | 39 (65.0) | 0.67* |
| | Primary (grade 1–5) | 4 (6.7) | 8 (13.3) | |
| | Secondary (grade 6–10) | 9 (15.0) | 9 (15.0) | |
| | Higher (grade >11) | 5 (8.3) | 4 (6.7) | |
| **Occupation** | | | | |
| | Employed | 22 (36.7) | 16 (26.7) | 0.478 |
| | Household work | 19 (31.7) | 27 (45.0) | |
| | Agriculture or labor | 14 (23.3) | 13 (21.7) | |
| | Unemployed | 5 (8.3) | 4 (6.7) | |
| **Economic status** | | | | |
| | Above poverty line | 49 (81.7) | 50 (83.3) | 0.81 |
| | Poor** | 11 (18.3) | 10 (16.7) | |

* Fischer's exact test,

**If family income per person per year was less than Rupees 16, 355 (US$: 163)

mmHg in usual care and 137.8 (17.0) mmHg in intervention. The mean difference in SBP among the intervention group was significantly higher than that in the control group (p<0.041). Similarly, the mean (SD) of DBP was 98.8 (13.2) mmHg in baseline and 93.3 (13.7) mmHg during follow-up in the control group. The mean (SD) of DBP reduced from 100.41 (14.5) mmHg to 89.46 (9.4) mmHg after intervention (Table 4 and Fig 2).

The average change of SBP (SD) from follow-up to baseline was significantly higher in the intervention group with -18.7 (19.8) mmHg than in the control group at -11.2 (20.1) mmHg (p = 0.041). A similar change in DBP (SD) was -5.53 (17.1) mmHg in usual care and -10.95 (14.6) mmHg after intervention (Table 4 and Fig 2). In a subgroup analysis of gender, mean difference of SBP was significantly higher in the intervention group compared to the control among women (p = 0.01) and age cohort of 45 to 59 years (p = 0.033). Similarly, a difference in

**Table 2. Baseline smoking, alcohol consumption, fruit and vegetable intake, physical activities and drug intake.**

| Variables | | Control (n = 60) | Intervention (n = 60) | P value |
|---|---|---|---|---|
| | | n (%) | n (%) | |
| **Smoking** | | | | |
| | Yes | 13 (21.7) | 8 (13.3) | 0.476 |
| | No | 47 (78.4) | 52 (86.7) | |
| **Alcohol consumption** | | | | |
| | Yes | 13 (21.7) | 22 (36.7) | 0.071 |
| | No | 47 (78.3) | 38 (63.3) | |
| **Inadequate Fruit and Vegetable intake** | | | | |
| | Yes | 59 (98.3) | 56 (93.3) | 0.171 |
| | No | 1 (1.7) | 4 (6.7) | |
| **Level of physical activities** | | | | |
| | Low | 18 (30.0) | 28 (46.7) | 0.06 |
| | Active | 42 (70.0) | 32 (53.3) | |
| **Proportion of hypertension treatment** | | | | |
| | Yes | 32 (53.3) | 24 (40.0) | 0.143 |
| | No | 28 (46.7) | 36 (60.0) | |

DBP was significantly higher in the intervention group among participants aged 60 and above (p = 0.049) (S5 Table).

The total knowledge score (SD) increased from 4.46 (1.6) to 6.85 (1.8) in the intervention group. The mean difference (follow-up minus baseline) of this score was 2.38 (2.4) in the intervention arm, which was significantly higher than that of the control group, 0.13 (1.8) (p<0.001) (Table 4). Linear regression analysis showed knowledge score mean difference rose significantly (S1 Table). The mean change (SD) in BMI (follow-up minus baseline) was -0.18 (2.31) and -0.71 (2.1) in control and intervention, respectively. The mean difference (SD) of waist circumference was -0.23 (9.1) cm in the usual care and -0.73 (8.2) cm in the intervention group (Table 4).

## Sensitivity analysis

On sensitivity analysis of all 125 data including imputed missing value, unadjusted odds ratio (95% CI) was 2.1 (1.03–4.29). The intervention was significantly effective (p = 0.041) in controlling SBP, as seen in the main analysis. The odds ratio (95% CI) of controlled DBP and controlled BP were 1.52 (0.75–3.14) and 1.86 (0.87–3.93), respectively. As in the main analysis,

**Table 3. Intervention effects on outcome variables.**

| Variables | Baseline (n = 120) | | Follow-up (n = 120) | | Unadjusted Odds Ratio | | Adjusted Odds Ratio | |
|---|---|---|---|---|---|---|---|---|
| | Control (n = 60), n (%) | Intervention (n = 60), n (%) | Control (n = 60), n (%) | Intervention(n = 60), n (%) | OR (95% CI) | P value | OR (95% CI) | P value |
| **Controlled SBP** | 2 (3.3) | 2 (3.3) | 24 (40.0) | 35 (58.3) | 2.1 (1.01–4.35) | **0.046** | 2.94[a] (1.25–6.9) | **0.013** |
| **Controlled DBP** | 11 (18.3) | 9 (15.0) | 23 (38.3) | 27 (45.0) | 1.31 (0.63–2.72) | 0.459 | 1.24[b] (0.55–2.80) | 0.600 |
| **Controlled BP** | 0.00 | 0.00 | 17 (28.3) | 23 (38.3) | 1.57 (0.73–3.38) | 0.247 | 1.79[c] (0.74–4.35) | 0.194 |

a Adjusted for clustering effect, age, sex, ethnicity, level of education, baseline mean SBP and total knowledge score

b Adjusted for clustering effect, age, sex, ethnicity, level of education, baseline mean DBP and total knowledge score

c Adjusted for clustering effect, age, sex, ethnicity, level of education, baseline mean SBP, DBP and total knowledge score

**Table 4. Mean blood pressure, body mass index, waist circumference, and knowledge score related to hypertension.**

| Variables | | Control (n = 60) | | Intervention (n = 60) | | P value (difference of means) |
|---|---|---|---|---|---|---|
| | | Mean (SD) | P value | Mean (SD) | P value | |
| **Systolic Blood Pressure** | | | | | | |
| | Baseline | 152.1 (12.9) | < .001 | 156.6 (17.4) | < .001 | |
| | Follow-up | 140.9 (19.5) | | 137.8 (17.0) | | |
| | Change (Follow-up minus baseline) | -11.2 (20.1) | | -18.7 (19.8) | | **0.041** |
| **Diastolic Blood Pressure** | | | | | | |
| | Baseline | 98.8 (13.2) | **0.015** | 100.41 (14.5) | < .001 | |
| | Follow-up | 93.3 (13.7) | | 89.46 (9.4) | | |
| | Change (Follow-up minus baseline) | -5.53 (17.1) | | -10.95 (14.6) | | 0.065 |
| **Total Knowledge Score** | | | | | | |
| | Baseline | 4.05 (1.5) | 0.572 | 4.46 (1.6) | < .001 | |
| | Follow-up | 4.1 (1.2) | | 6.85 (1.8) | | |
| | Change (Follow-up minus baseline) | 0.13 (1.8) | | 2.38 (2.4) | | **<0.001** |
| **Body Mass Index** | | | | | | |
| | Baseline | 26.13 (3.95) | 0.527 | 24.63 (3.9) | **0.011** | |
| | Follow-up | 25.94 (3.9) | | 23.91 (3.5) | | |
| | Change (Follow-up minus baseline) | -0.18 (2.31) | | -0.71 (2.1) | | 0.197 |
| **Waist Circumference** | | | | | | |
| | Baseline | 95.33 (13.6) | 0.844 | 93.43 (11.6) | 0.495 | |
| | Follow-up | 95.10 (12.08) | | 92.70 (11.1) | | |
| | Change (Follow-up minus baseline) | -0.23 (9.1) | | -0.73 (8.2) | | 0.754 |

these were non-significant. After adjusting for age, sex, clustering effect, ethnicity, level of education, and baseline SBP or DBP, or both, results were almost similar to the main analysis (Table 5).

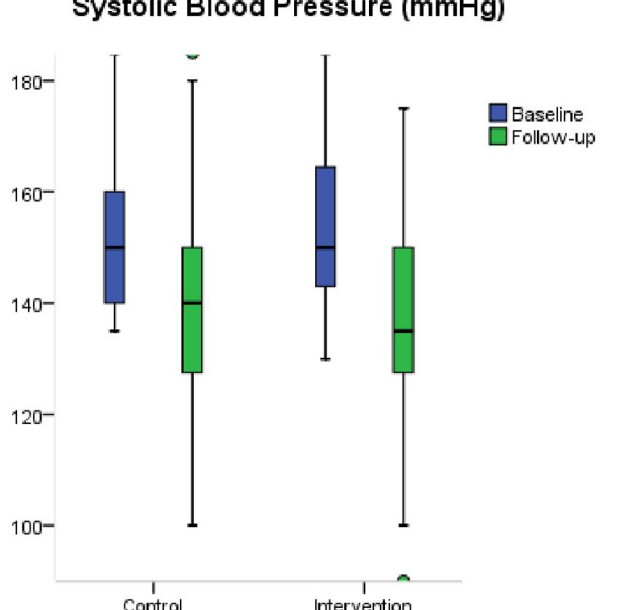
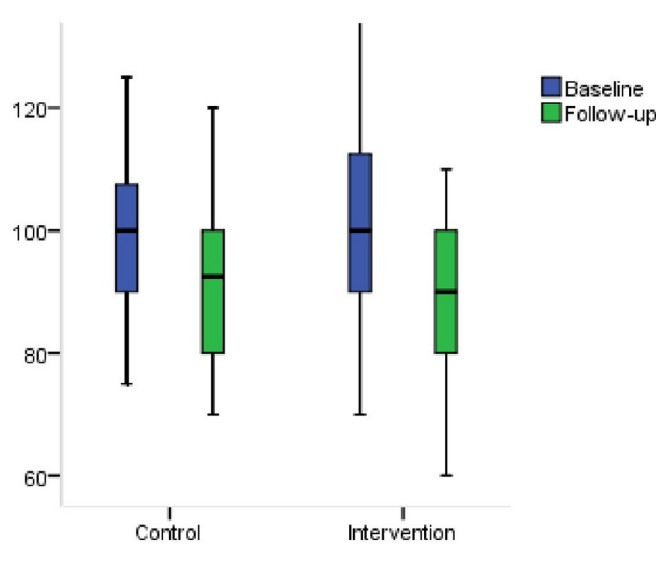

**Fig 2. Box-plot of mean SBP and DBP of intervention and control group.**

**Table 5. Sensitivity analysis of intervention effects on outcome variables.**

| Variables | Baseline (n = 125) | | Follow-up (n = 125) | | Unadjusted Odds Ratio | | Adjusted Odds Ratio | |
|---|---|---|---|---|---|---|---|---|
| | Control (n = 62) n (%) | Intervention (n = 63) n (%) | Control (n = 62) n (%) | Intervention (n = 63) n (%) | OR (95% C.I.) | P value | OR (95% C.I.) | P value |
| Controlled SBP | 2 (3.2) | 2 (3.2) | 26 (41.9) | 38 (60.3) | 2.1 (1.03–4.29) | 0.041 | 2.84[a] (1.26–6.42) | 0.012 |
| Controlled DBP | 11 (11.7) | 10 (15.9) | 23 (37.1) | 30 (47.6) | 1.52 (0.75–3.14) | 0.235 | 1.63[b] (0.75–3.56) | 0.213 |
| Controlled BP | 0.00 | 0.00 | 17 (27.4) | 26 (41.3) | 1.86 (0.87–3.93) | 0.105 | 2.26[c] (0.97–5.23) | 0.056 |

a Adjusted for clustering effect, age, sex, ethnicity, level of education, and baseline mean SBP.

b Adjusted for clustering effect, age, sex, ethnicity, level of education, and baseline mean DBP.

c Adjusted for clustering effect, age, sex, ethnicity, level of education, and baseline mean SBP, DBP.

## Discussion

Our study demonstrated that health education related to blood pressure by health professionals in a small group along with community support is effective to attain a normal range of blood pressure in community settings. This study provides evidence that health education and home visits can be used to the existing health systems of low-and-middle-income countries such as Nepal. This helps to maintain satisfactory blood pressure control by increasing knowledge and awareness among patients with hypertension.

Our findings are in line with other similar studies conducted in various settings. One similar cluster randomized controlled trial conducted in rural India reported that three months of health education intervention reduced blood pressure by 20.3% among the intervention group compared to 9.5% among usual care [25]. Another study conducted in Argentina showed that community health worker-led home-based intervention implemented for six months improved the proportion of those with controlled BP from 17.0% at baseline to 45.6% at 6 months in the intervention group compared to 17.6.% at baseline to 40.8% in the control group [26]. Our study also revealed that the change in systolic and diastolic blood pressure (follow-up minus baseline) after health education and home support was higher among the intervention group compared to the control group. Our findings are consistent with that of the COBIN trial [27], a recent community based clustered randomized controlled trial in Nepal. This trial showed that the mean SBP decreased by 6.47 mmHg after one year in the health education and lifestyle intervention group and 2.85 mmHg in the control group and the mean DBP changed by 2.9 mmHg compared to 1.11 mmHg, respectively, among the hypertensive patients [27]. Our results also correspond to the data released by Gamang et. al. [25] from a study in India, which showed the SBP decreased by 8.2 mmHg in health education intervention in comparison to 2.1 mmHg in the control group. DBP was also reduced by 4.2 mmHg and 2.2 mmHg, respectively, in intervention and control groups [25].

Our results also showed that the knowledge score was significantly higher among the participants who received health education. These insights and learning about hypertension might have helped to reduce blood pressure among the intervention group. It can be expected that improved knowledge might significantly affect adherence to healthy lifestyle and regular use of medication. Many studies demonstrated the link between knowledge score and adoption and maintenance of a healthy lifestyle. For instance, one meta-analysis of 13 studies conducted in middle-income countries showed that group education and other supportive methods improved the adherence to lifestyle modification [28]. Similarly, another study revealed group counselling and home visits encourage the participants to adopt healthy self-care practices [29]. Therefore, health education is a tool to improve patient engagement and to begin and maintain adherence to lifestyle modification and compliance with medications [30] which is

ultimately one of the strongest tools for the prevention of cardiovascular disease [31]. The long-term consequence of health literacy among patients of chronic illness is that it may prevent co-morbidities [32].

Our study generated evidence of utilizing health education related to blood pressure in our existing health system to control hypertension in low-and-middle-income countries such as Nepal. Nepal is constantly developing its health system. Majority of wards (small administrative units) of urban and rural municipalities have at least one health institution such as health post or urban health clinics [33]. We can mobilize currently existing resources for hypertension and healthy lifestyle education as we had conducted in our study. Group counselling could be conducted for hypertensive patients in those health institutions by currently working health professionals along with community support by Female Community Health Volunteers (FCHV). This can have a significant impact on blood pressure control and benefit on other diseases such as obesity, diabetes and cardiovascular risk factors. Neupane et al. [27] reported that utilizing the existing FCHV of Nepal can significantly reduce blood pressure level in community settings among normotensive, prehypertensive and hypertensive patients. This concept has also been studied in China and India, which found that health belief model-based health education could significantly improve the blood pressure control of hypertensive patients in the community [25, 34].

The current study has several strengths. We believe this to be the first study that conducted both group counselling and frequent home visits for uncontrolled hypertensive patients in community settings of Nepal. This study was unique in its design in Nepal, where we applied clustered randomization assuming whole ward (small administrative unit) as a clustered. Furthermore, we designed the syllabus for the group counselling and explained the study steps in detail to facilitate the replication. Additionally, our study had a comparison with the usual care group for comparison of the effects of intervention. We had also used different data enumerators for baseline and follow-up data collection to minimize the assessor's bias.

However, our study holds some limitations. At the outset, we could use only a small sample size. This implies the result must be interpreted carefully. Furthermore, as during the first phase of the study, we randomly screened the hypertensive patients only in four clusters [8], we could not increase the number of clusters for the trial. We could not perform the ancillary analysis for the effect of health education on lifestyle modification and antihypertensive drug adherence for these two reasons. In addition, as it was a six-month trial, we could not minimize the seasonal variation of blood pressure by taking blood pressure measurements in the same season [35, 36]. Apart from this, we had provided the syllabus to the medical graduates and registered nurses, but we did not conduct training sessions for them. They taught the participants with the best of their academic knowledge. Because of this, information sharing might have been varied between the groups. Finally, as we conducted our study only in a single municipality, the findings should be cautiously generalized to other settings.

## Conclusions

In conclusion, multiple sessions of group health education complemented by frequent household visits by trained health volunteers can effectively improve the hypertension related knowledge and reduce blood pressure among people with uncontrolled hypertension at the community level in Nepal. A large-scale research with a larger sample size is needed to assess whether this approach can be cost-effective and the findings can be generalized for a larger population in Nepal.

## Supporting information

**S1 Table. Syllabus for health education.**
(DOCX)

**S2 Table. Linear regression model for change in knowledge score.**
(DOCX)

**S3 Table. CONSORT extension for cluster trials 2010 checklist.**
(DOCX)

**S4 Table. Trial protocol.**
(DOC)

**S5 Table. Distribution of mean difference at 6-month follow-up SBP and DBP by gender, age, in both control and intervention group.**
(DOCX)

## Acknowledgments

This study was supported by Birendranagar municipality and Bheri Zonal Ayurveda Hospital. The authors want to acknowledge the staff of Bheri Zonal Ayurveda Hospital, Surkhet for providing logistic and managerial support. We cannot forget the help from Batabaran Bhawan and Bhairav Secondary School for providing a venue to conduct health education sessions. Similarly, they would like to express especial thanks to field supervisor and data collector team: Ms. Nirmala, Ms Lata, Mr. Narendra, Mr. Tulsi, Mr. Man Bahadur, Ms.Deepa, Ms. Bishnu, Ms. Debisara, Ms. Laxmi, Ms. Kalpana, Ms. Shamjhana, Ms. Prema, Ms. Chitra, Ms. Sita, Ms. Jwoti, Ms. Shobha, Ms. Ranjana, Ms. Pratiksha, Ms. Amrita, Mr. Deepak. The study team wants to thank Chair Person of ward number 4 of Birendranagar Municipality Mr. Nilkantha Khanal and Female Community Health Volunteers (FCHV) of ward number 4, 5, 8 and 10 for supporting in the community mobilization. In the end, authors want to acknowledge Royal Australasian College of Physicians for supporting Career Development Fellowship to B.d.C.

## Author Contributions

**Conceptualization:** Mahesh Kumar Khanal, Pratiksha Bhandari, Raja Ram Dhungana, Pratik Bhandari, Lal B. Rawal, Yadav Gurung, K. N. Paudel, Amit Singh, Surya Devkota, Barbora de Courten.

**Data curation:** Mahesh Kumar Khanal, Pratiksha Bhandari, Raja Ram Dhungana, Pratik Bhandari, Yadav Gurung.

**Formal analysis:** Mahesh Kumar Khanal, Pratiksha Bhandari, Raja Ram Dhungana, Lal B. Rawal, Amit Singh, Surya Devkota, Barbora de Courten.

**Funding acquisition:** Yadav Gurung.

**Investigation:** Mahesh Kumar Khanal, Pratiksha Bhandari, Pratik Bhandari, Lal B. Rawal, K. N. Paudel, Amit Singh.

**Methodology:** Mahesh Kumar Khanal, Raja Ram Dhungana, Yadav Gurung, K. N. Paudel, Amit Singh.

**Project administration:** Pratiksha Bhandari, Raja Ram Dhungana, Yadav Gurung, K. N. Paudel, Amit Singh.

**Resources:** Pratiksha Bhandari, Pratik Bhandari, Yadav Gurung, Surya Devkota.

**Software:** Raja Ram Dhungana.

**Supervision:** Mahesh Kumar Khanal, Yadav Gurung, K. N. Paudel, Amit Singh.

**Validation:** Pratiksha Bhandari, K. N. Paudel, Surya Devkota.

**Visualization:** Pratik Bhandari, Lal B. Rawal, Barbora de Courten.

**Writing – original draft:** Mahesh Kumar Khanal.

**Writing – review & editing:** Mahesh Kumar Khanal, Pratiksha Bhandari, Raja Ram Dhungana, Pratik Bhandari, Lal B. Rawal, Yadav Gurung, K. N. Paudel, Amit Singh, Surya Devkota, Barbora de Courten.

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
