## [Decision Letter · Decision Letter 0]

13 Aug 2021

PONE-D-21-20934

Effectiveness of community-based health education and home support program to reduce blood pressure among the patients with uncontrolled hypertension in Nepal: A cluster-randomized trial

PLOS ONE

Dear Dr. Khanal,

Thank you for submitting your manuscript to PLOS ONE. After careful consideration, we feel that it has merit but does not fully meet PLOS ONE’s publication criteria as it currently stands. Therefore, we invite you to submit a revised version of the manuscript that addresses the points raised during the review process.

We look forward to receiving your revised manuscript.

Kind regards,

Yoshihiro Fukumoto

Academic Editor

PLOS ONE

Journal Requirements:

Reviewers' comments:

Reviewer's Responses to Questions

**Comments to the Author**

1. Is the manuscript technically sound, and do the data support the conclusions?

Reviewer #1: Partly

Reviewer #2: Yes

Reviewer #3: Yes

2. Has the statistical analysis been performed appropriately and rigorously? 

Reviewer #1: No

Reviewer #2: Yes

Reviewer #3: Yes

3. Have the authors made all data underlying the findings in their manuscript fully available?

Reviewer #1: No

Reviewer #2: Yes

Reviewer #3: Yes

4. Is the manuscript presented in an intelligible fashion and written in standard English?

Reviewer #1: No

Reviewer #2: Yes

Reviewer #3: Yes

5. Review Comments to the Author

Reviewer #1: Authors reported that multiple health education sessions complemented by frequent household visits by health volunteers can effectively improve knowledge on hypertension and reduce the blood pressure among uncontrolled hypertensive patients at community level in Nepal. This manuscript may be interesting. However, there are several concerns in this manuscript.

Major comments:

1) The most serious concern may be a small number of subjects as authors raised it in the section of limitation. The small sample size will lead a wrong conclusion.

2) In Tables 3 and 6, authors should not analyze data by adjusting for SBP and DBP simultaneously.

3) In Tables 3-5, how do authors deal with the use of antihypertensive medication, which may impact on systolic and diastolic blood pressures?

4) The section of “Results” is too long and redundant.

5) Authors should ask natives to edit English.

Minor comments:

1) Although systolic blood pressure and diastolic blood pressure were already abbreviated as SBP and DBP, authors were using systolic blood pressure and diastolic blood pressure without abbreviation again and again.

2) In the footnote of Table 1, Fischar exact test may be wrong. Fisher’s exact test may be correct. Authors should add the explanation in the section of data analysis.

3) In the section of result, Effectiveness of intervention: line 312; “After adjusting for” is correct. Line 316; “we adjusted for” is correct.

4) In the section of result, Sensitivity analysis: not “Odds Ratio” but “odds ratio.” Line 353; “After adjusting for” is correct.

5) In the section of Discussion, line 380: not “8.2m Hg” but “8.2mmHg”.

6) The journal names of the references should be accurately abbreviated.

Reviewer #2: Major comments:

This study was a research article which was showing that the multiple health education sessions can improve knowledge on hypertension and reduce the blood pressure among uncontrolled hypertensive patients. The concept of the present study was valuable but there are some concerns as described below.

1. As it was unclear the improvements of parameters such as SBP, DBP and the knowledge score after interventions, authors should better describe the change of parameters as “follow-up minus baseline”.

2. The authors didn’t satisfyingly describe why the multiple health education programs were effective to reduce blood pressure among uncontrolled hypertension patients. Authors should reveal that why systolic blood pressure was improved among uncontrolled hypertension patients after intervention, for example the reduction of salt, their physical activity or the antihypertensive drug adherence?

Minor comments:

1. Page 3; line 42-44: As the mean change in SBP calculated follow-up minus baseline, “-18.7mmHg vs. -11.2mmHg”, “-10.95mmHg vs. -5.53mmHg” are correct.

2. Page 5; line 58: not “10・8”but ”10.8”

3. Figure 1: The excluded numbers in participate are needed to add in figure 1 and the section of “participants’ flow”.

4. Page 19; Table 4: not “baseline and follow-up” but “follow-up minus baseline”, and authors should correct values on table 4 and manuscripts.

5. Page 23; line 376-382: not “6・47, 2・85” but “6.47, 2.85”, not “mm Hg” but “mmHg”.

Reviewer #3: A cluster-randomized controlled trial was conducted which aimed, primarily to compare the proportion of participants with controlled systolic blood pressure (SBP) between the intervention and control arms. The proportion with controlled SBP was significantly higher in the intervention arm compared to the control arm.

Minor revisions:

1- Line 115: Typographical error: Alpha

2- Line 251: Specify the descriptive statistics used to summarize the data.

3- Line 253: An underlying assumption of the chi-square test is independence of samples. The baseline and follow-up outcome variables are not independent; therefore, the chi-square test is inappropriate for comparing these outcomes.

4- Line 262: State the numbers of data points requiring imputation.

5- Table 1 Footnote:

a. Fisher’s is misspelled.

b. For clarity, associate double asterisks with the second comment.

6. PLOS authors have the option to publish the peer review history of their article (what does this mean?). If published, this will include your full peer review and any attached files.

Reviewer #1: No

Reviewer #2: No

Reviewer #3: No

---

## [Author Response · Author response to Decision Letter 0]

24 Aug 2021

I have attached separate files to address the issues raised by reviewers.

---

## [Decision Letter · Decision Letter 1]

31 Aug 2021

PONE-D-21-20934R1

Effectiveness of community-based health education and home support program to reduce blood pressure among the patients with uncontrolled hypertension in Nepal: A cluster-randomized trial

PLOS ONE

Dear Dr. Khanal,

Thank you for submitting your manuscript to PLOS ONE. After careful consideration, we feel that it has merit but does not fully meet PLOS ONE’s publication criteria as it currently stands. Therefore, we invite you to submit a revised version of the manuscript that addresses the points raised during the review process.

We look forward to receiving your revised manuscript.

Kind regards,

Yoshihiro Fukumoto

Academic Editor

PLOS ONE

Journal Requirements:

Reviewers' comments:

Reviewer's Responses to Questions

**Comments to the Author**

1. If the authors have adequately addressed your comments raised in a previous round of review and you feel that this manuscript is now acceptable for publication, you may indicate that here to bypass the “Comments to the Author” section, enter your conflict of interest statement in the “Confidential to Editor” section, and submit your "Accept" recommendation.

Reviewer #1: All comments have been addressed

Reviewer #2: (No Response)

Reviewer #3: All comments have been addressed

2. Is the manuscript technically sound, and do the data support the conclusions?

Reviewer #1: Yes

Reviewer #2: Yes

Reviewer #3: (No Response)

3. Has the statistical analysis been performed appropriately and rigorously? 

Reviewer #1: Yes

Reviewer #2: Yes

Reviewer #3: (No Response)

4. Have the authors made all data underlying the findings in their manuscript fully available?

Reviewer #1: Yes

Reviewer #2: Yes

Reviewer #3: (No Response)

5. Is the manuscript presented in an intelligible fashion and written in standard English?

Reviewer #1: Yes

Reviewer #2: Yes

Reviewer #3: (No Response)

6. Review Comments to the Author

Reviewer #1: The revised manuscript has been much improved according to the reviewers' helpful comments.

This reviewer does not have any further comments.

Reviewer #2: Minor comments:

A few comments were not addressed.

1. Page 5; line 58: not “10・8” but ”10.8”

2. Page 24; line 392-398: not “6・47, 2・85” but “6.47, 2.85”

3. Page 24; line 393,394, 397, 398: not “mm-Hg” but “mmHg”.

Reviewer #3: (No Response)

7. PLOS authors have the option to publish the peer review history of their article (what does this mean?). If published, this will include your full peer review and any attached files.

Reviewer #1: No

Reviewer #2: No

Reviewer #3: No

---

## [Author Response · Author response to Decision Letter 1]

1 Sep 2021

I have addressed issues raised by reviewer. It is uploaded as a separate file.

---

## [Decision Letter · Decision Letter 2]

27 Sep 2021

Effectiveness of community-based health education and home support program to reduce blood pressure among patients with uncontrolled hypertension in Nepal: A cluster-randomized trial

PONE-D-21-20934R2

Dear Dr. Khanal,

We’re pleased to inform you that your manuscript has been judged scientifically suitable for publication and will be formally accepted for publication once it meets all outstanding technical requirements.

Kind regards,

Yoshihiro Fukumoto

Academic Editor

PLOS ONE

Additional Editor Comments (optional):

Reviewers' comments:

Reviewer's Responses to Questions

**Comments to the Author**

1. If the authors have adequately addressed your comments raised in a previous round of review and you feel that this manuscript is now acceptable for publication, you may indicate that here to bypass the “Comments to the Author” section, enter your conflict of interest statement in the “Confidential to Editor” section, and submit your "Accept" recommendation.

Reviewer #2: All comments have been addressed

2. Is the manuscript technically sound, and do the data support the conclusions?

Reviewer #2: Yes

3. Has the statistical analysis been performed appropriately and rigorously? 

Reviewer #2: Yes

4. Have the authors made all data underlying the findings in their manuscript fully available?

Reviewer #2: Yes

5. Is the manuscript presented in an intelligible fashion and written in standard English?

Reviewer #2: Yes

6. Review Comments to the Author

Reviewer #2: (No Response)

7. PLOS authors have the option to publish the peer review history of their article (what does this mean?). If published, this will include your full peer review and any attached files.

Reviewer #2: No

---

## [Editor Report · Acceptance letter]

4 Oct 2021

PONE-D-21-20934R2 

Effectiveness of community-based health education and home support program to reduce blood pressure among patients with uncontrolled hypertension in Nepal: A cluster-randomized trial 

Dear Dr. Khanal:

I'm pleased to inform you that your manuscript has been deemed suitable for publication in PLOS ONE. Congratulations! Your manuscript is now with our production department. 

Kind regards, 

on behalf of

Dr. Yoshihiro Fukumoto 

Academic Editor

PLOS ONE